# Environmental effects on explosive detection threshold of domestic dogs

**Lauren S. Fernandez**[1], **Sarah A. Kane**[1], **Mallory T. DeChant**[1],
**Paola A. Prada-Tiedemann**[2], **Nathaniel J. Hall**[1]*

1 Canine Olfaction Research and Education Lab, Davis College of Animal and Food Science, Texas Tech University, Lubbock, Texas, United States of America, 2 Forensic Analytical Chemistry and Odor Profiling Lab at Texas Tech University, Lubbock, Texas, United States of America

* Nathaniel.J.Hall@ttu.edu

**Data Availability Statement:** All data is fully available without restriction. This is provided in supplemental information. We have provided all temperature data, olfactometer data, and the R

## Abstract

Detection canines are deployed to detect explosives in a wide range of environmental conditions. These environmental conditions may have negative impacts on canine capabilities as a sensor. This study leveraged an air dilution olfactometer to present controlled odor concentrations of four different energetic materials (double base smokeless powder, Composition C4, ammonium nitrate, and flake Trinitrotoluene) to dogs working in a range of high temperature, standard, and low temperature conditions with high and low humidity conditions. The air dilution olfactometer controlled concentrations independent of environmental condition. Dogs' detection threshold limits were measured using a descending staircase procedure. We measured dogs' threshold twice for each energetic under each environmental condition. Results indicated heterogeneity in effects based on energetic, but all odors were detected at their lowest concentrations under standard conditions. Smokeless powder detection was reduced under all environmental conditions compared to standard and was least detectable under high temperature and humidity conditions. AN detection was poorest under high temperature high and low humidity conditions. C4 in contrast, was least detectable at low temperatures with high humidity. TNT detection was difficult under all conditions, so decrements due to environmental conditions were not statistically detectable. Additional measures were also found to be associated with detection limits. Under high temperature conditions, correlations were observed between canine mean subcutaneous temperature and detection limits, such that dogs experiencing greater temperature increases showed poorer detection limits. In addition, dog's latency to sample the odor port from the onset of a trial was longest in the high temperature conditions. Longer latencies were also predictive of poorer detection performance. Overall, dogs showed deficits in detection sensitivity limits under all environmental conditions for at least one energetic material when the concentration of that energetic material was not directly impacted by the environmental conditions. These results suggest that behavioral factors related to environmental exposure can have important impacts on canine detection sensitivity and should be considered in operational environments.

code used for analysis. We have also provided the behavioral coding results for PRE.

**Funding:** This research was made possible through funding provided by the DoD Army Research Office under Contract No. W911NF2120124. https://www.arl.army.mil/who-we-are/aro/. SAK's work was supported by the National Science Foundation Graduate Research Fellowship Program (DGE 2140745).https://www.nsfgrfp.org/. This funder played no role in the study design, data collection, analysis, preparation or decision to publish this manuscript.

**Competing interests:** The authors have declared that no competing interests exist.

## Introduction

Dogs have a highly developed olfactory system giving them keen odor detection capabilities. Humans have used this ability in the dog for scent detection tasks, including detection of diseases [1, 2], pathogens [3], human scent [4], cancer [5–7], narcotics [8], and explosives [9–11]. For this reason, military, law enforcement, and other government agencies heavily rely on the detection capabilities of working dogs.

For decades, explosives detection dogs (EDD) have been one of the main tools for the detection of energetic materials and explosives [12]. The remarkable capabilities of these dogs, including their mobility and sensitivity, make it possible to detect various munitions used in acts of terrorism or war in a timely and efficient manner. Previous studies have successfully shown that dogs can be trained to detect a wide variety of energetic materials with excellent sensitivity and specificity [13].

It is important, however, to acknowledge the limitations of dogs' capabilities to detect specific concentrations of relevant target odorants. Studies have been conducted to quantify threshold detection limits of working dogs to various substances. For example, accelerant detection canines can detect low concentrations of gasoline, as low as 0.1μL [14, 15]. Dogs' detection threshold for methyl benzoate, an odor associated with cocaine, has also been measured to be approximately 16ppb [16]. Two dogs' detection threshold for amyl acetate was shown to be 1.14 ppt and 1.90 ppt [17], several folds lower than that found for methyl benzoate, highlighting that detection limits vary substantially by odorant.

Quantifying canine detection limits is an important research task, however, there has been few studies on canine thresholds with energetic materials. This can be important, especially for energetics, because many have very low vapor pressure, indicating limited availability of the energetic molecule as an odor source [18].

In addition to the vapor pressure of the odor, environmental conditions can also physically affect the dog and odor source [19, 20]. Working dogs may need to be deployed in environments that reach temperatures higher than 40°C and lower than 0°C [21, 22]. Environmental conditions can have important implication for canine odor detection [19] and dogs' general working ability [22]. Extreme conditions such as these pose a threat of heat exhaustion or hypothermia, potentially impeding the dog's ability to perform odor detection work. The standard range of canine temperature core body temperature is between 37.2°C and 39.2°C [22]; previous studies have demonstrated that while working, dogs can exceed 40.6°C [23]. Temperatures of this severity have been shown to increase fatigue and cause clinical signs of heat exhaustion [22, 24]. Gazit and Terkel (2003) showed that increases in panting decreased detection accuracy [25]. This suggests that hot conditions could lead to poorer olfactory sensitivity in the dog.

The humidity of the environment can also influence heat stress risk [26, 27]. During strenuous work or activity in extreme conditions, heat from the core of the dog is transferred to the skin via passive conduction through the tissue, and cooling is achieved by blood flow to the skin through active convection [27]. During this time the dog will begin to pant in attempt to thermoregulate. This process is drastically impeded as the moisture content in the air increases [27]. Previous studies have found that dogs working in extreme hot and humid environmental conditions display greater heat stress at a quicker rate [28].

Due to heat stress, there is a potential that detection dogs may experience a decrease in detection accuracy while working in hot and humid environments. It has been found that landmine detection dogs had a significant decrease in detection accuracy after the working environment experienced heavy rain, ultimately increasing the humidity [28]. However, it should also be considered that the heavy rain potentially disrupted the soil containing the odor

by washing the odorant away. Further data in this study showed that dogs had a decrease in detection accuracy early in the morning when the humidity was the highest and had an increase in detection success as humidity decreased throughout the day [28]. However, a literature review by Jenkins et al., 2018 noted that a moist environment is crucial for olfactory perception, indicating that humidity may aid in olfaction [29].

Cold stress and hypothermia are additional potential environmental concerns for working dogs. Diverio et al., (2016) found that search and rescue dogs working in -8.5˚C to -10.4˚C showed acclimatization to the extreme environment conditions and completed the search task without showing signs of stress or fatigue. This search task, however, only lasted 10 minutes, which is not representative of the duration of time a dog may be tasked with working in the cold [30]. It is necessary to establish changes in detection accuracy for dogs working in cold environments for longer durations. A survey conducted in 2012 found that out of 53 handlers working with avalanche dogs 39% reported a decrease in dog performance while working in -23˚C, -12˚C [31] Previous work investigating changes in detection performance of methyl benzoate in various environmental conditions, found dogs had the lowest threshold while working in cold and standard conditions compared to high temperature environments [20]. Cold temperature effects have not been investigated in explosive detection work.

Given the limited data evaluating canine olfactory detection performance to energetic materials in a variety of operationally relevant environmental conditions, the goal of this project was to create a systematic evaluation of temperature and humidity effects on canine detection for a series of energetic materials. Few prior studies accounted for how odor availability may impact canine performance. For this reason, we kept the energetic materials at a controlled environmental condition and manipulated temperature (40˚C vs 0˚C) and humidity conditions (90/70% vs 40/50%) for dogs. We also measured canine physiological parameters during the detection task to better understand how these measures relate to detection performance.

## Methods and materials

### Participants

Participants included eight spayed/ neutered adult working dogs of two breeds (Refer to Table 1). All dogs were previously eliminated from working dog programs and had previous training and/or odor work experience. Detailed descriptions of each participant can be found in Table 1. Dogs varied in their reasons for removal from their prior working program including preference for food vs. toy rewards, distractibility or environmental sensitivities. Dogs received a minimum of two walks a day and obedience training to maintain enrichment and exercise during the duration of the study. The primary reinforcement for testing and training

**Table 1. Participant description.**

| Dog | Approximate Age (Years) | Breed | Approximate weight (kg) | Reproductive Status |
|---|---|---|---|---|
| Zulu | 4 | German Shorthaired pointer | 22.68 | Altered male |
| Luna | 2 | German Shorthaired pointer | 27.67 | Altered female |
| Moni | 3 | German Shorthaired pointer | 24.49 | Altered female |
| Dasty | 3 | Labrador retriever | 27.21 | Altered male |
| Jack | 2 | Labrador retriever | 29.48 | Altered male |
| Bello | 3 | German Shorthaired pointer | 25.63 | Altered male |
| Dalton | 2 | Labrador retriever | 34.02 | Altered male |
| Rocket | 2 | Labrador retriever | 27.21 | Altered male |

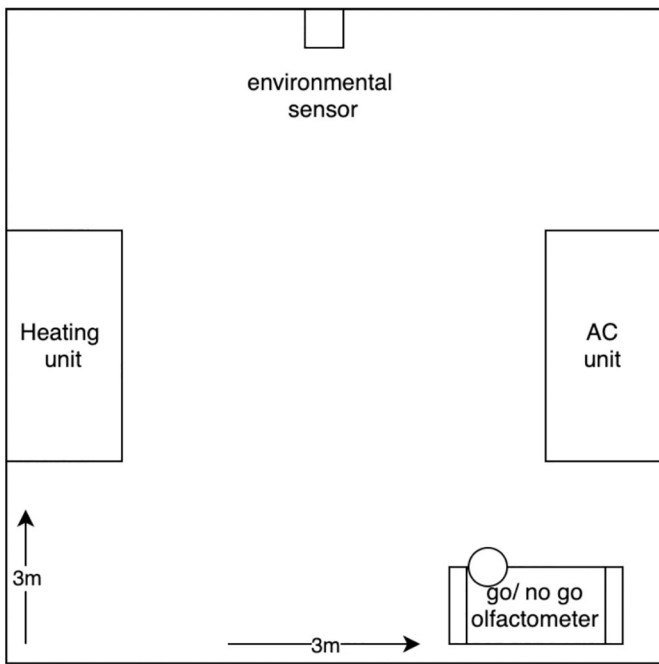

**Fig 1. Schematic diagram of environmental chamber.** Testing environment set up, containing a built-in heater and AC unit. An environmental sensor was attached to the back wall to monitor the environmental conditions while testing to maintain proper conditions. A go/ no go olfactometer testing apparatus was placed in the front of the room to preform training and testing.

were dog treats and dog food. Therefore, one-quarter of the total food ration was fed to the dogs in the morning, and the remaining 75% was given as a second meal at the end of the day. Dogs received additional food throughout the day during walks and training. Lastly, all dogs had free access to water at all times.

**Animal welfare considerations.** All procedures utilized in the following experiments were approved by the Texas Tech University Institutional Animal Care and Use Committee (Protocol # 21051–07) and approved by the US Army Medical Research and Development Command Animal Care and Use Office (#78018-ST-H.e001). During testing in environmental conditions, participants were monitored to avoid hypothermia and hyperthermia. All dogs were first trained to detect odors at 85% accuracy or higher, while in standard environmental conditions. Thus, if the participant showed signs of distress or refusal to search for five consecutive trials, then the dog was immediately removed from the chamber and the session was terminated. Furthermore, dogs were tested for a maximum of 40 minutes, after which dogs were removed from the chamber, and sessions were terminated.

**Experimental set-up.** An environmental chamber measuring 3m x 3m was the primary testing and training environment during this study (see Fig 1). The chamber had built in Heating, Ventilation, Air Conditioning (HVAC) and supplemental heater units to allow the room to reach temperatures below 0˚C and above 43˚C. Supplemental equipment, such as dehumidifiers and humidifiers, were placed in the room to control relative humidity conditions.

## Odorants

Four energetic materials were used in this experiment for testing and training. Double-base smokeless powder (SP), prill ammonium nitrate (AN), flaked trinitrotoluene (TNT), and

**Table 2. Description of odorants.**

| Energetic material | Mass (g) | Total flow rate (L/min) | Temperature of water bath (˚C) | Brand of energetic material | Vial size Diameter (cm) |
|---|---|---|---|---|---|
| Double-base smokeless powder | 10g | 10 L/min | 30˚C | Hodgdon H335 | 2.54 cm |
| Ammonium nitrate | 20g | 3 L/min | 40˚C | Omni explosives | 5.08 cm |
| Trinitrotoluene | 10.5g | 3 L/min | 40˚C | Omni explosives | 5.08 cm |
| C4 | 8g | 3 L/min | 40˚C | Omni explosives | 2.54 cm |

Composition C4 (C4) were used in this experiment (see Table 2). Materials were supplied by an energetics supplier (see Table 2) at the start of the experiment. Each material, in a glass vial, was placed into a heated water bath during training and testing to standardize odor environmental conditions. Because odor availability is dependent on temperature, the temperature of the water bath was based on the odor identity and was increased for less volatile odors to improve detection. All odors underwent systematic air dilutions during testing for baseline and experimental environmental conditions. Due to expected variation in canine detection capability, the flow rate used for dilution was dependent on odor (Table 2). Based on the olfactometer design, the overall flow rate determined the highest range of concentrations that could be presented. During initial training, if dogs struggled to detect the 0.01 dilution of any odor at >85% accuracy, the total flow was set to 3 SLPM to allow for a higher range of concentrations to be presented.

Samples of AN and TNT were placed in wider glass vials to increase the surface area of the material. Lastly, smokeless powder underwent air dilution factors of 0.01, 0.00316, 0.001, 0.000316, and 0.0001. The remaining energetics underwent dilution factors of 1.0, 0.316, 0.1, 0.0316, and 0.01, due to dogs failing to reach qualification criterion (>85% accuracy) at a dilution factor of 0.01. Note that final air flow of non-odor trials was the same as the final air flow of diluted odor trials.

### Data collection equipment

Participants were trained and tested using an automated Go/NoGo air dilution olfactometer system (Fig 2). The system utilized Alicat (Tucson, AZ, USA) gas mass airflow controllers to create a serial air dilution of the odorant being tested. A glass vial (containing the explosive) was placed in a water bath set at the desired temperature (30˚C– 40˚C) for each specific odor being tested. The glass vial was then pierced with two Teflon (PTFE) tubes to allow for clean air introduction and displacement of sample headspace into the system to begin a serial dilution. The odorant/ air mixture then travels through the mass airflow controllers and static mixers to create the desired air dilution. Lastly, the diluted air traveled to the port to allow participants to sniff the odor. The total airflow delivered to the dog was dependent on the odor material and ranged from 3 L/min to 10L/min but was always consistent for all dilutions for the same odor. Refer to Fig 2 for a detailed diagram of the function and air flow of the apparatus.

The olfactometer was designed for Go/NoGo testing and therefore delivered either clean air, or air containing volatiles of the target odors to a stainless-steel odor port. The probability for a correct response was 50%. The odor port was fitted with an exhaust fan, which exhausted odor from the odor port during an inter-trial intervals. The odor port also contained an infrared beam pair which detected canine nose entries into the port. Canine responses were measured by the infrared beam pair, in which dogs were trained to hold their nose in the port for 4 continuous seconds as a Go response. A NoGo response was scored when the dog sampled the odor port (inserting their nose) followed by the removal of their nose for 4 consecutive

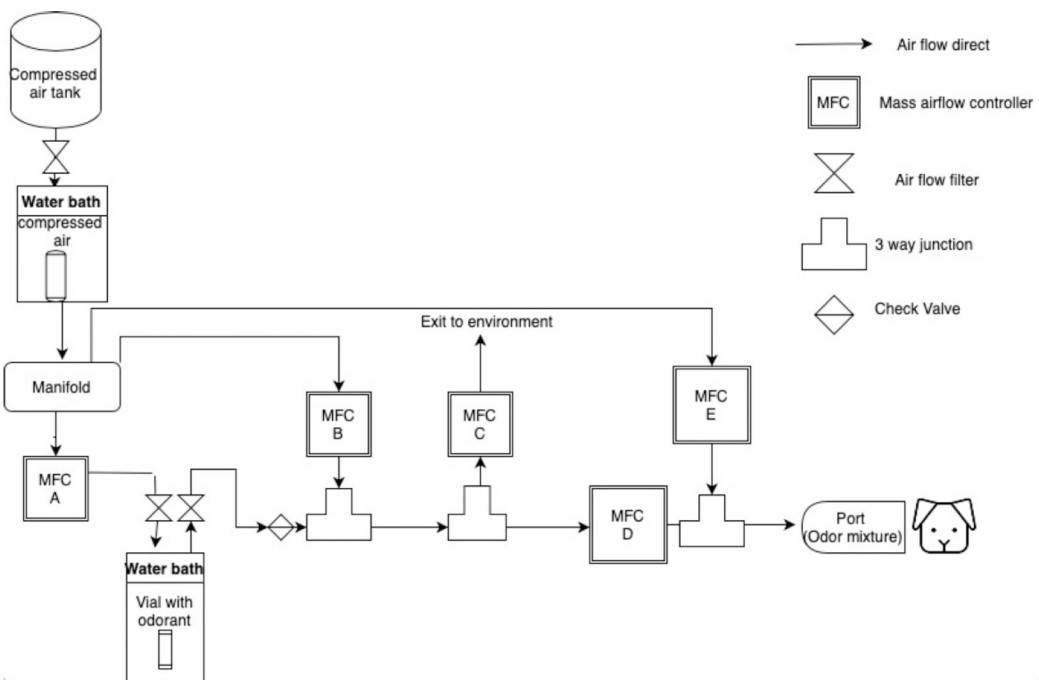

**Fig 2. Diagram of mass air flow dilution olfactometer.** The path of air flow is shown from left to right. Clean air travels through a warm water bath heated to replicate the same temperature of the odorant water bath. The clean air then travels to the manifold and is dispersed to the MFC A, MFC B and MFC E. Air from MFC A is pushed into the vial containing an odorant located in a separate water bath to collect the headspace from the vial and then pushed out of the vial to a three-way junction connected to MFC B. The air from the vial is mixed with clean air from the manifold the travels to the remaining mass air flow controllers. The odorant is systematically diluted by the mass air flow controllers and then pushed down the odor line to be delivered to the port.

seconds. Tones were used by the computer to indicate if a response was correct or incorrect to a handler that was blind to the presence/absence of the odor.

The correct tone was played when a dog responded with a four second nose hold in the port when the port contained the diluted target (hit), or if the dog responded by inserting their nose into the port and disengaging from the port when the port contained clear air, a non-target trial (correct rejection). Furthermore, an incorrect tone was played when participants incorrectly held their nose for four seconds in the port during a non-target trial (false alert) or disengaged from the port during a target present trial (miss). A "timeout" was recorded if the dog did not sample the odor port within 45s of a trial initiation tone and being prompted to search by the handler. Inter-trial intervals were 30 seconds. For the last 10 seconds of the inter-trial interval, the olfactometer presented the odor of the next trial to ensure a stable concentration was presented prior to the dog sampling.

## Physiological measures

**Heart rate.** Participants wore a Polar H10 heart rate monitor during each testing session. However, due to substantial missing data due to movement artifacts, heart rate was excluded from further statistical analysis.

**Subcutaneous temperature.** Subcutaneous temperature of each participant was taken before entering the chamber and every five minutes during testing. Temperature data was

collected by scanning a HomeAgain TempScan 134.2 kHz ISO microchip [32] located subcutaneously in the shoulder of the dog. The temperature was recorded by the handler. Microchips were implanted in the dogs while working in previous training facilities before entering the canine olfaction lab.

**Respiration effort (RE).** Videos were coded using BORIS [33] to determine the changes of the dogs' respiratory effort throughout the duration of the session and differences between conditions. Each video was coded at five-minute intervals (0, 5, 10, 15, 20, etc.) for the first five seconds of each interval. The first five second clip started at the start of the first trial. However, for the video to meet criteria for coding the dog needed to be in sight of the camera with 75% of the body showing, to determine if the abdomen and chest were actively involved in the respiratory process. Further, the dog's nose needed to be outside of the odor port and the mouth needed to be visible. If a video clip did not meet criteria, the video was scanned for the nearest five second clip that met this criteria from the specific interval. A RE scale rating system described by DeChant et al. (submitted) was used to determine the respiration effort of each participant during the five second clip for each five-minute interval. Scoring for this scale ranges from 0 to 10 (Refer to S1 Table). Zero being defined as resting and breathing is an involuntary process, and ten being very heavy breathing and the chest and abdomen are expanding and collapsing violently. For inter-observer agreement, two coders involved in the data collection scored the same 20% of the videos for RE. Inter-class correlations were used to assess agreement.

## Environmental conditions

Canine's detection sensitivity was tested twice for each energetic (4 odors) in four different extreme environmental conditions and at a standard condition. Details of the environmental conditions can be found in Table 3. Conditions were selected to match possible extremes encountered in an operational setting. A sensor (SensorPush HT.w wireless thermometer/hygrometer sensor; rated accuracy: ± 2% RH and ± 0.2˚C) was placed in the room that monitored temperature and humidity conditions throughout the testing session to ensure the environmental conditions remained within limits of variation.

## Canine odor detection training and testing

**Preliminary training.** Dogs were previously trained to three explosive compounds (double-base smokeless powder, AN, and C4) using an automated 3-alternative line-up olfactometer while at another training facility. Previous research provides a detailed description of the automated odor delivery system used for initial training [34]. Once dogs arrived at the Canine Olfaction Lab they were then initially trained to TNT using a Go/NoGo air dilution system.

Following initial training, dogs were transferred to the Go/NoGo air dilution system, consisting of one port presenting either clean air (non-target trials) or the odor dilution of the

**Table 3. Environmental conditions of threshold assessment.**

| Condition | Temperature ˚C (± 5 degrees) | Relative humidity (±5%) |
|---|---|---|
| Baseline | 21˚C | 50% |
| High temperature/ high humidity | 40˚C | 70% |
| High temperature/ low humidity | 40˚C | 40% |
| Low temperature/ high humidity | 0˚C | 90% |
| Low temperature/ low humidity | 0˚C | 50% |

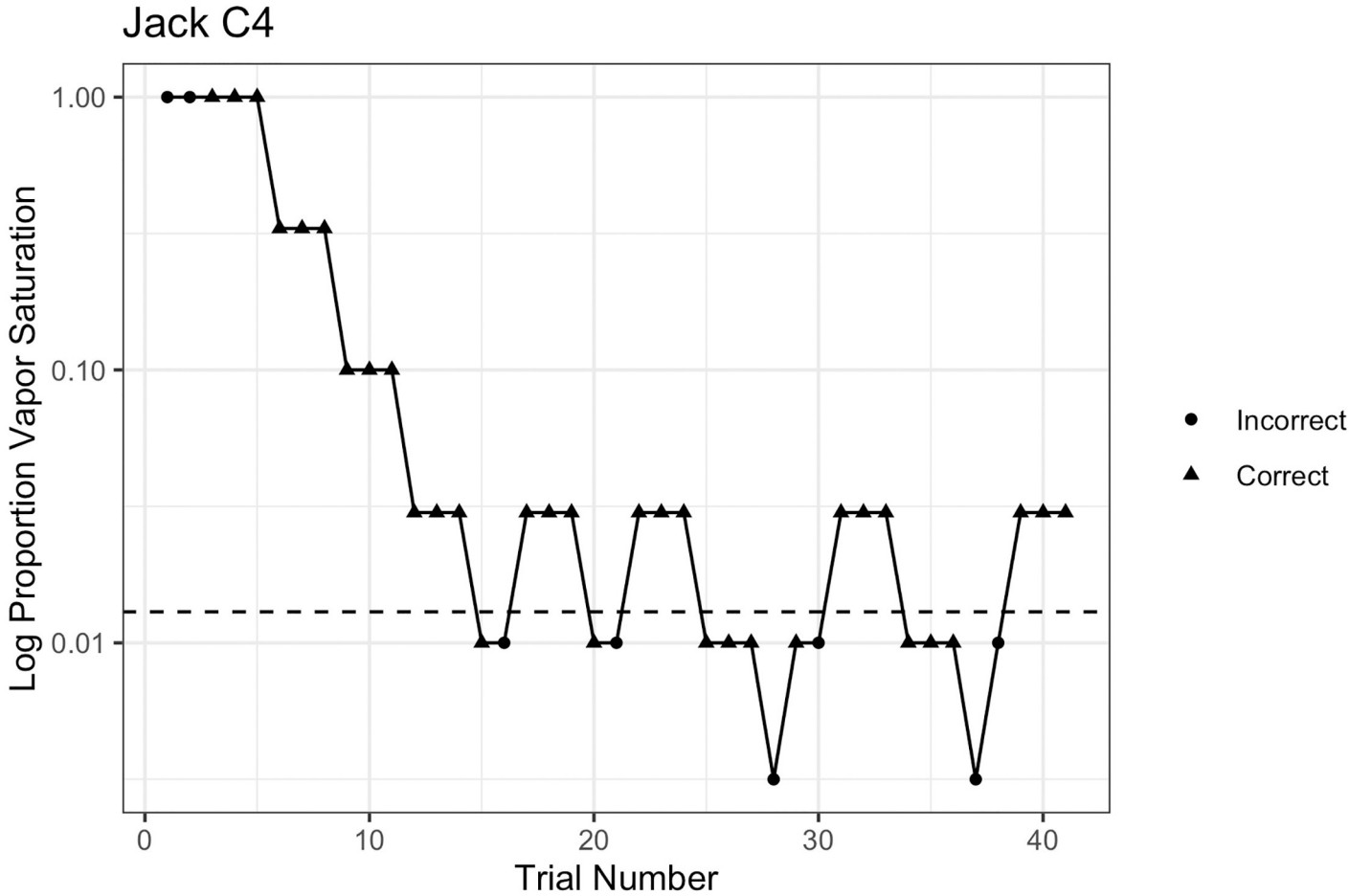

**Fig 3. The dog Jack following the 3-down 1-up adaptive threshold procedure for C-4 in standard conditions.** After three correct responses, the concentration was decreased by a half-log. Once Jack responded incorrectly when presented with 0.01 dilution of C4, the concentration was increased by a half-log. Jack correctly alerted 0.01 dilution of C4 but was not able to detect lower concentrations. Eight reversals were reached during testing and threshold was calculated as the geometric mean of the last six reversals.

specific energetic being tested (50% probability of odor presence). Each dog received two sessions of 20 trials with each odor of the four odors or the necessary number of sessions needed to achieve 85% accuracy or higher for two consecutive double blinded sessions.

**Canine odor threshold testing.** Once participants reached initial training criteria, dogs were tested on the Go/NoGo air dilution system with a descending staircase, three-down one-up procedure. Dogs were blindly presented with both odor and non-odor trials during testing (equal probability, such that at least one of each trial type occurred every three trials). Following the descending staircase method, if the dog made three correct consecutive responses, the concentration of the odor was automatically lowered by a half-log step (see Fig 3). The concentration decreased until the dog made an incorrect response. If incorrect, the concentration was raised to the previous dilution step. Concentration was lowered or raised over the session until dogs showed 8 reversals in the direction of change of concentration. The session continued until dogs either reached 8 reversals, 40 min of training, or met a welfare criterion for discontinuing (described in Animal Welfare Conditions).

### Experimental design

Dogs completed all testing for one odorant before proceeding to the next in the following order (SP, C4, AN, TNT). For each odor, dogs completed initial Go/NoGo qualification training. After meeting qualification, dogs completed two threshold assessments at standard conditions. Dogs were then pseudo-randomly assigned to one of two conditions such that half of the dogs started in the hot and humid conditions working to hot/dry, cold/dry, and ended with cold/humid. The other half started in the cold/humid conditions, then moved to cold/dry, hot/dry, and ended with hot/humid. Dogs completed two threshold sessions per environmental condition before advancing to the next environmental condition. Dogs alternated between starting in hot or cold conditions when advancing from one target odorant to the next.

### Statistical analysis

Threshold was calculated as the geometric mean of the last 6 (of 8) reversal points. If a dog did not complete 8 reversals due to early termination (due to welfare criterion, exceeding the session time, or failing to achieve three consecutive responses to decrease in concentration) the starting concentration (i.e., highest concentration) was imputed as the missed reversal points to reflect poorer performance.

To evaluate the impact of environmental condition a linear mixed effect model was fit in which log transformed threshold was predicted by the odor, environmental condition, and their interaction. A random intercept was fit for each dog. Fixed effects were evaluated with Anova from the car package [35]. False discovery rate adjusted post hoc tests were conducted in which each environmental condition was compared to standard conditions.

The R code is available in Supplemental information.

## Results

### Threshold

Average thresholds across all eight dogs for each odor in each condition are shown in Fig 4, which highlights large differences in sensitivity limits for the different energetics. Dogs showed highest sensitivity for SP followed by C4, AN and TNT. A linear mixed effect model showed a main effect of odor type on threshold detection limits ($X_2$ = 2,512, df = 3, p<0.001). Post hoc tests indicate there was a significant difference in sensitivity between all energetics except for between AN and TNT (t = 0.83, p = 0.84).

Table 4 shows the detection limits (log transformation of proportion of vapor saturation) for each target odor under environmental conditions and standard conditions. A mixed effect model indicated that log threshold was unrelated to breed ($X_2$ = 0.11, df = 1, p<0.74). However, there was a significant odor by condition interaction ($X_2$ = 38.45, df = 12, p<0.001) as well as main effects of odor ($X_2$ = 2,922, df = 3, p<0.001) and condition ($X_2$ = 28.02, df = 4, p<0.001) on the log threshold scale.

To evaluate the odor by condition interaction, post hoc tests were conducted between conditions for each energetic. All environmental conditions were compared to the standard environmental condition (room temperature) as the control (see Table 4). Each environmental condition led to a decrement in detection limits for at least one energetic material. Importantly, performance was not better in an environmental condition in comparison to the standard condition. High Temp & High Humidity, High Temp & Low Humidity, and Low Temp & High Humidity all led to decrements for two target odors. Low Temp & Low humidity only impacted smokeless powder, and overall had the smallest magnitude impact on performance (See Fig 5 and Table 5). Further, examination of individual canine performance in all

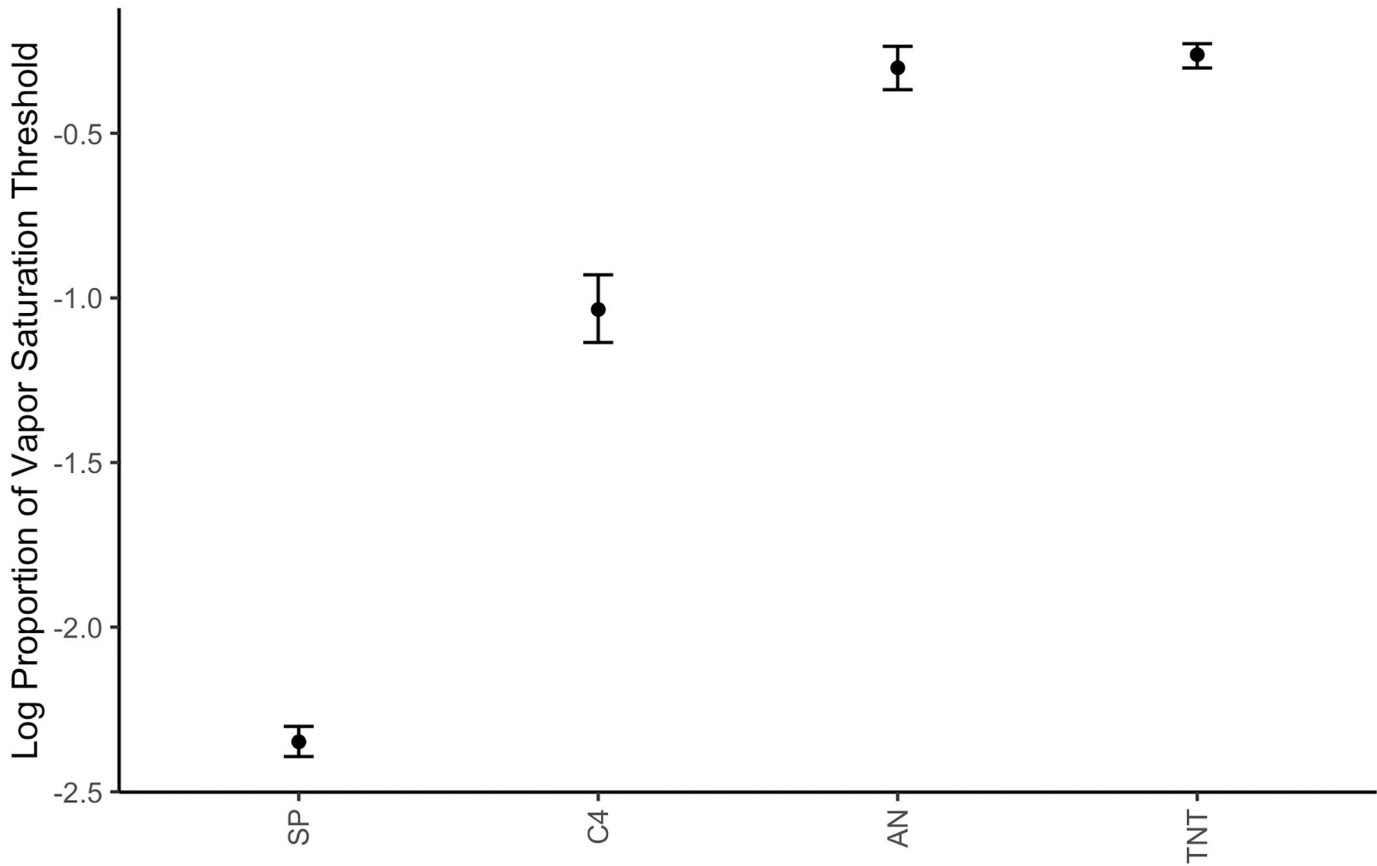

**Fig 4. Average threshold of each odor.** Displaying average dilution of vapor saturation threshold across all dogs for both testing sessions in standard conditions of each odor. SP, C4, AN, and TNT. Error bars show 95% bootstrap estimated confidence intervals.

conditions detecting SP showed that individual patterns followed the mean trends (Refer to Fig 6). Six out of eight dogs showed the largest decrements in high temperature and high humidity and the least decrement in low temperature and low humidity.

Table 5 shows the post hoc results for each energetic material comparing detection threshold under each extreme condition to the standard condition. Positive t-ratio values indicate higher (poorer) detection threshold in the extreme environment compared to standard. Detection threshold for AN was poorer (compared to standard) at high temperature high humidity

**Table 4. Detection limit (log of vapor saturation) for each odor in each condition and 95% confidence intervals.**

| Odor | | Standard | High Temp High Humid | High Temp Low Humid | Low Temp High Humid | Low Temp Low Humid |
|---|---|---|---|---|---|---|
| AN | Mean | -0.4511 | -0.2321 | -0.0907 | -0.2686 | -0.4636 |
| | 95% CI | -0.612, -0.2897 | -0.393, -0.0707 | -0.252, 0.0707 | -0.430, -0.1072 | -0.625, -0.3023 |
| C-4 | Mean | -1.1848 | -1.2181 | -0.9844 | -0.8170 | -0.9716 |
| | 95% CI | -1.346, -1.0235 | -1.380, -1.0568 | -1.146, -0.8230 | -0.978, -0.6556 | -1.133, -0.8102 |
| SP | Mean | -2.5939 | -2.1563 | -2.3230 | -2.2918 | -2.3751 |
| | 95% CI | -2.755, -2.4325 | -2.318, -1.9950 | -2.484, -2.1617 | -2.453, -2.1304 | -2.536, -2.2137 |
| TNT | Mean | -0.2515 | -0.2126 | -0.2114 | -0.2786 | -0.3524 |
| | 95% CI | -0.413, -0.0902 | -0.374, -0.0512 | -0.373, -0.0501 | -0.440, -0.1172 | -0.514, -0.1911 |

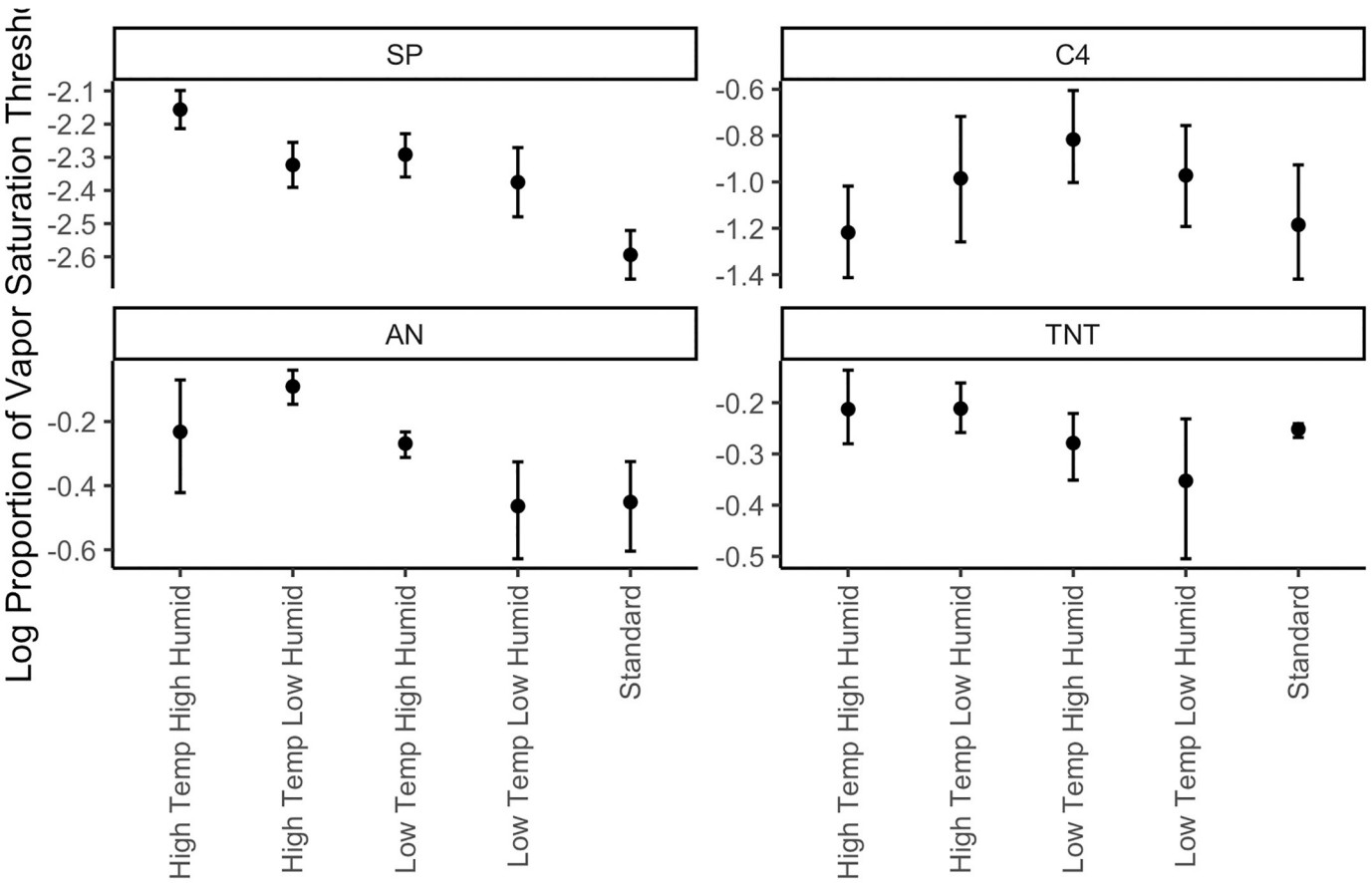

**Fig 5. Threshold (log proportion of vapor saturation) for each environmental condition and odor.** Error bars show bootstrap estimated 95% confidence intervals.

and high temperature low humidity. Comparisons for low temperature conditions did not reach statistical significance. C4 had poorer detection limits in low temperature high humidity and a trend (p = 0.06) for poorer detection at high temperature low humidity and low temperature low humidity. SP had poorer detection in all conditions, with the poorest detection limit in high temperature high humidity. TNT did not reach statistical significance for any comparison.

**Table 5. Threshold difference for each environment condition compared to standard condition.** Positive t-ratio indicates poorer sensitivity compared to standard.

| Conditions | High Temp High Humid vs. Standard | | High Temp Low Humid vs. Standard | | Low Temp High Humid vs. Standard | | Low Temp Low Humid vs. Standard | |
|---|---|---|---|---|---|---|---|---|
| Odor: | t-ratio | p | t-ratio | p | t-ratio | p | t-ratio | p |
| AN | 2.21 | 0.05 | 3.64 | <0.001 | 1.84 | 0.09 | -0.12 | 0.90 |
| C-4 | -0.03 | 0.73 | 2.02 | 0.06 | 3.71 | <0.01 | 2.15 | 0.06 |
| SP | 4.42 | <0.01 | 2.73 | 0.02 | 3.05 | <0.01 | 2.21 | 0.03 |
| TNT | 0.39 | 0.78 | 0.40 | 0.78 | -0.27 | 0.78 | -1.02 | 0.78 |

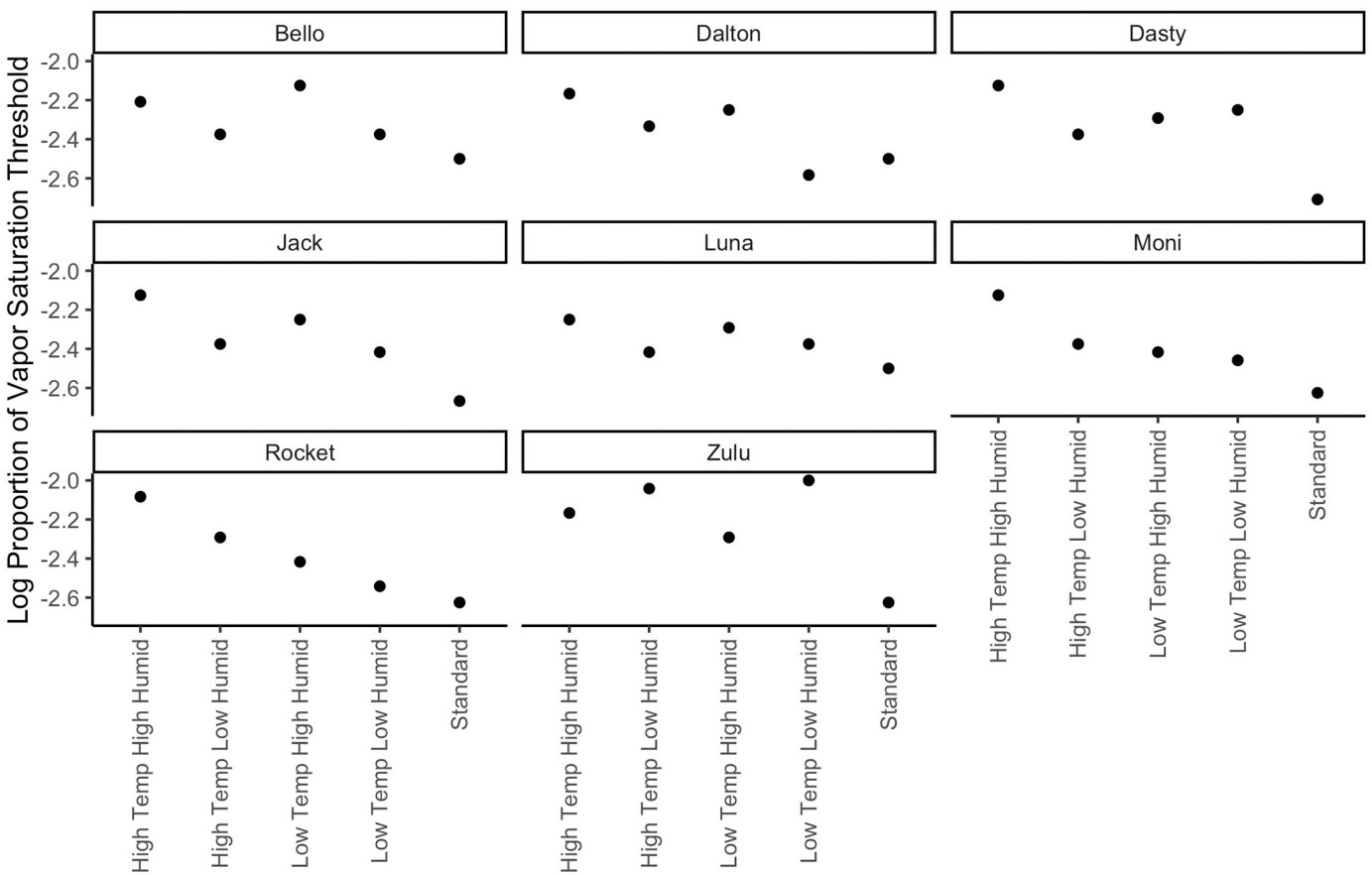

**Fig 6. Individual threshold difference for each environment condition compared to standard condition for dogs detecting SP.** Average threshold between both testing sessions for each condition.

### Subcutaneous temperature

Fig 7 shows the subcutaneous temperature across the first 20 minutes of the session. A cutoff time of 20 minutes was selected because nearly all sessions lasted this long. In the hot and standard conditions, subcutaneous temperature increased as dogs completed the threshold test. During the cold conditions, temperature decreased across sessions to below biologically plausible values, indicating the cutaneous sensor was susceptible to the room temperature conditions. Therefore, analyses were limited to standard and high temperature conditions.

First, a linear mixed effect model analysis of differences in mean subcutaneous temperatures between the three conditions indicated there was a significant difference in mean temperature between conditions ($X_2 = 242$, df = 2, p<0.001). Post hoc tests indicate that subcutaneous temperature was highest in High Temperature High Humidity in comparison to High Temperature Low Humidity (p<0.001) and Standard conditions (p<0.001), indicating an important impact of humidity between the two high temperature conditions at the same temperature. Further, High Temperature Low Humidity also led to higher subcutaneous temperatures compared to standard (p<0.001).

To evaluate the impact of mean subcutaneous temperature on threshold, a linear mixed effect model was fit in which threshold was predicted by odor (energetic), mean subcutaneous temperature and their interaction. There was a significant interaction between energetic and

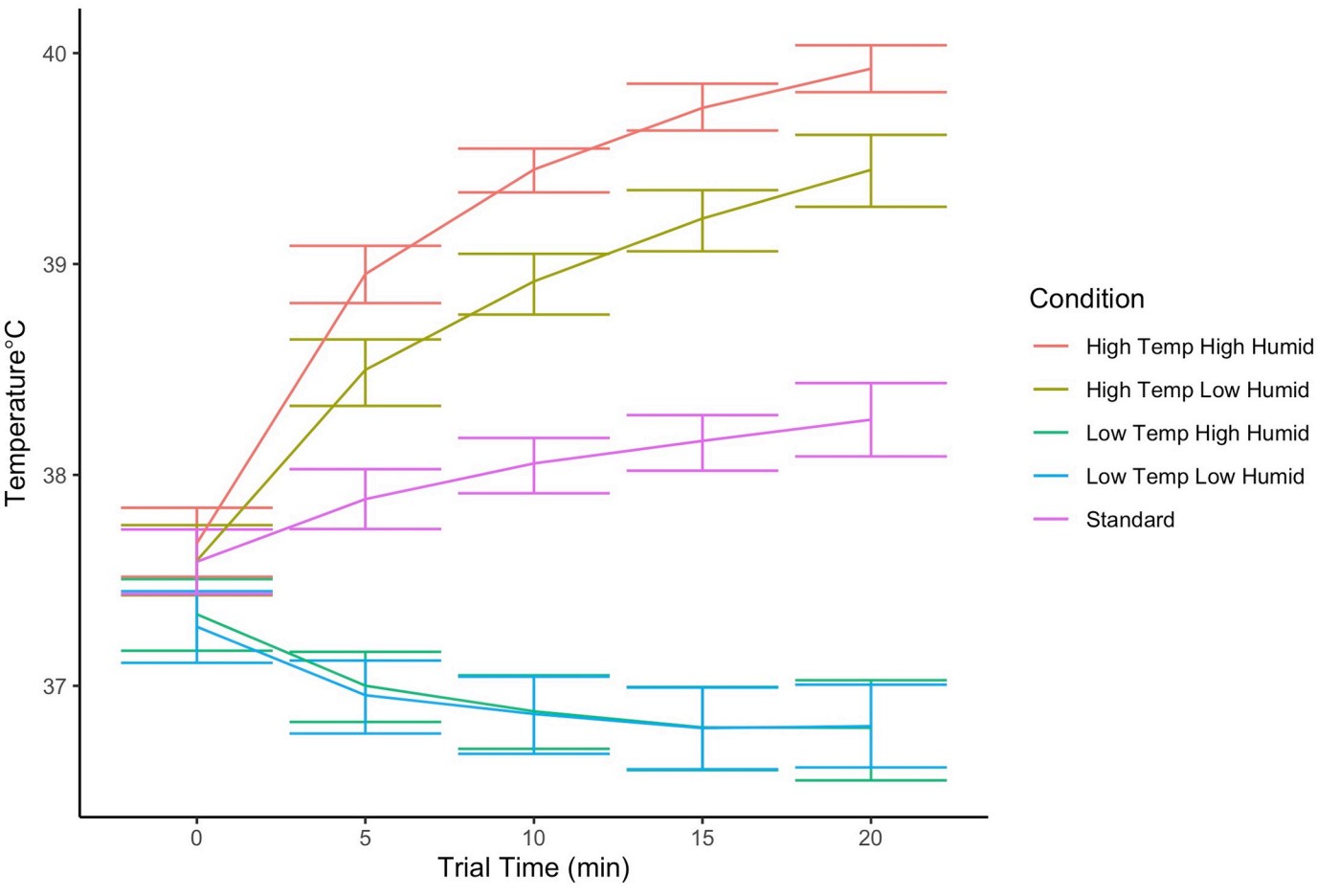

**Fig 7. Implanted skin temperature during testing.** Error bars show the 95% confidence interval.

mean temperature ($X_2$ = 8.64, df = 3, p = 0.03). To evaluate this interaction, a regression model was fit for each energetic. There was a significant positive relationship between mean subcutaneous temperature and threshold from SP ($X_2$ = 58.12, df = 1, p<0.001) and AN ($X_2$ = 7.16, df = 1, p = 0.007), but not for TNT or C4 (p>0.05; see Fig 8). This indicates that higher mean subcutaneous temperature was associated with lower sensitivity (poorer threshold) for SP and AN.

### Respiratory effort

Dogs showed increased RE score across the session during standard and hot conditions (high and low humidity) but stable scores in the low temperature conditions (see Fig 9). Linear mixed effect model shows that RE differed between the environmental conditions ($X_2$ = 948, df = 4, p<0.001). Post hoc tests indicate that there was a significant difference in RE score between all conditions except for the two low temperature conditions (t = 0.38, p = 0.99).

To evaluate whether RE was related to detection threshold, the log transformed threshold was predicted by odor (energetic), RE score and their interaction. There was a significant interaction ($X_2$ = 17.45, df = 3, p<0.001) between RE and odor/energetic on threshold. Conducting separate analysis for each energetic indicated that increased RE led to higher threshold

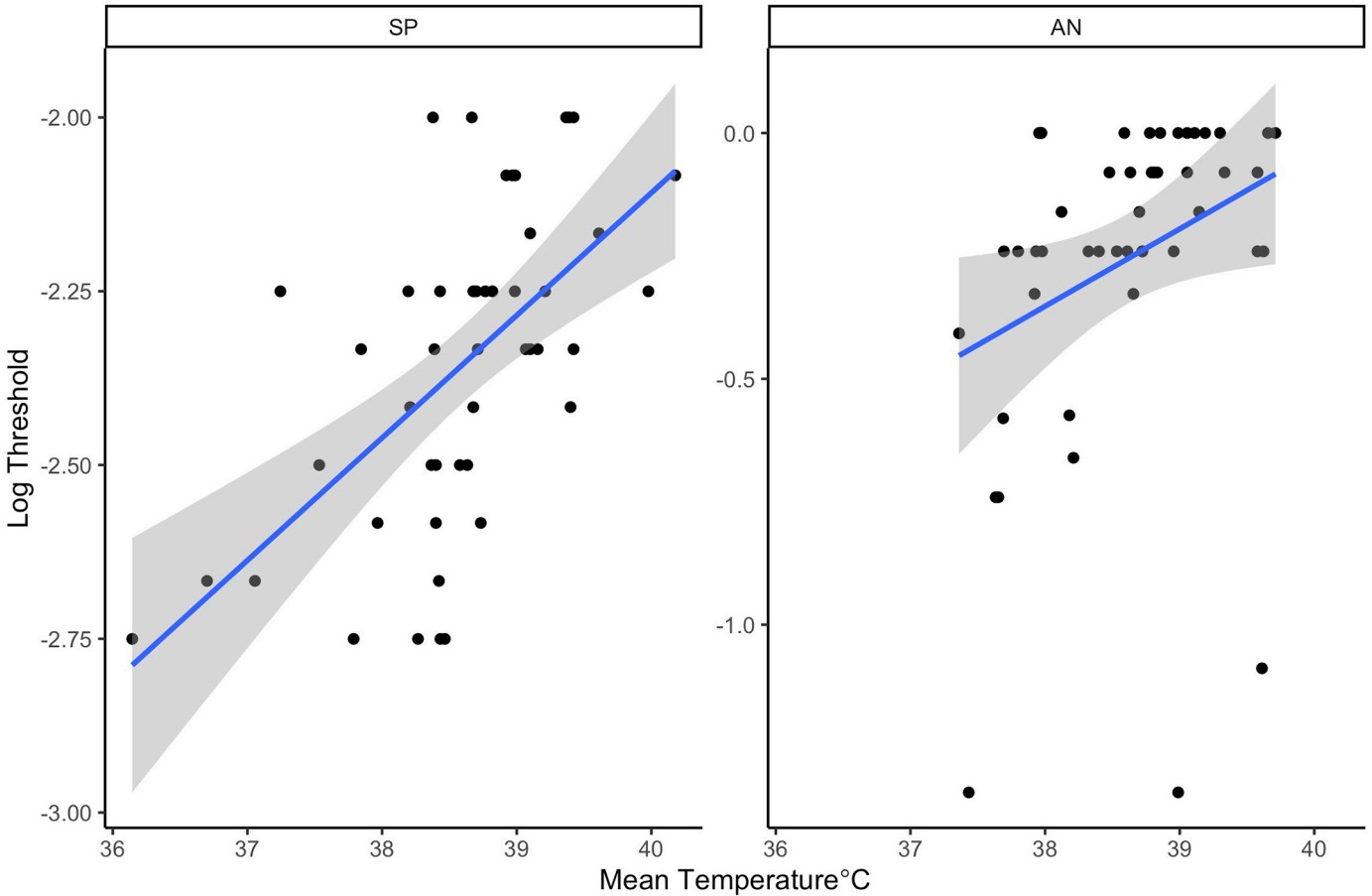

**Fig 8. Threshold relation to mean subcutaneous temperature.** Line shows the best fit regression.

for AN ($X_2$ = 4.71, df = 1, p<0.03), but for lower threshold for C4 ($X_2$ = 7.28, df = 1, p<0.01; see Fig 10). No significant effect of RE was found for SP ($X_2$ = 1.86, df = 1, p = 0.17) and TNT ($X_2$ = 2.67, df = 1, p = 0.1).

## Latency

Latency was scored as the time between the olfactometer initializing the trial with the start tone and the first nose poke into the odor port. Latency to initialize a trial varied by environmental condition ($X_2$ = 30.1, df = 4, p<0.001). Fig 11 and post-hoc tests shows the latency was highest in the two high temperature conditions but was similar in the low temperature and standard conditions.

A mixed effect model was fit in which log threshold was predicted by the mean session latency, energetic and their interaction. There was no significant interaction ($X_2$ = 1.85, df = 3, p = 0.60), but there was a main effect of energetic ($X_2$ = 2,762, df = 3, p<0.001) and latency ($X_2$ = 14.26, df = 1, p<0.001). Fig 12 shows there was a positive relationship such that increases in latency (or delay in initiating a trial) was associated with higher (or poorer) thresholds. Thus, when dogs quickly approached the odor port at the trial start, detection sensitivity was higher.

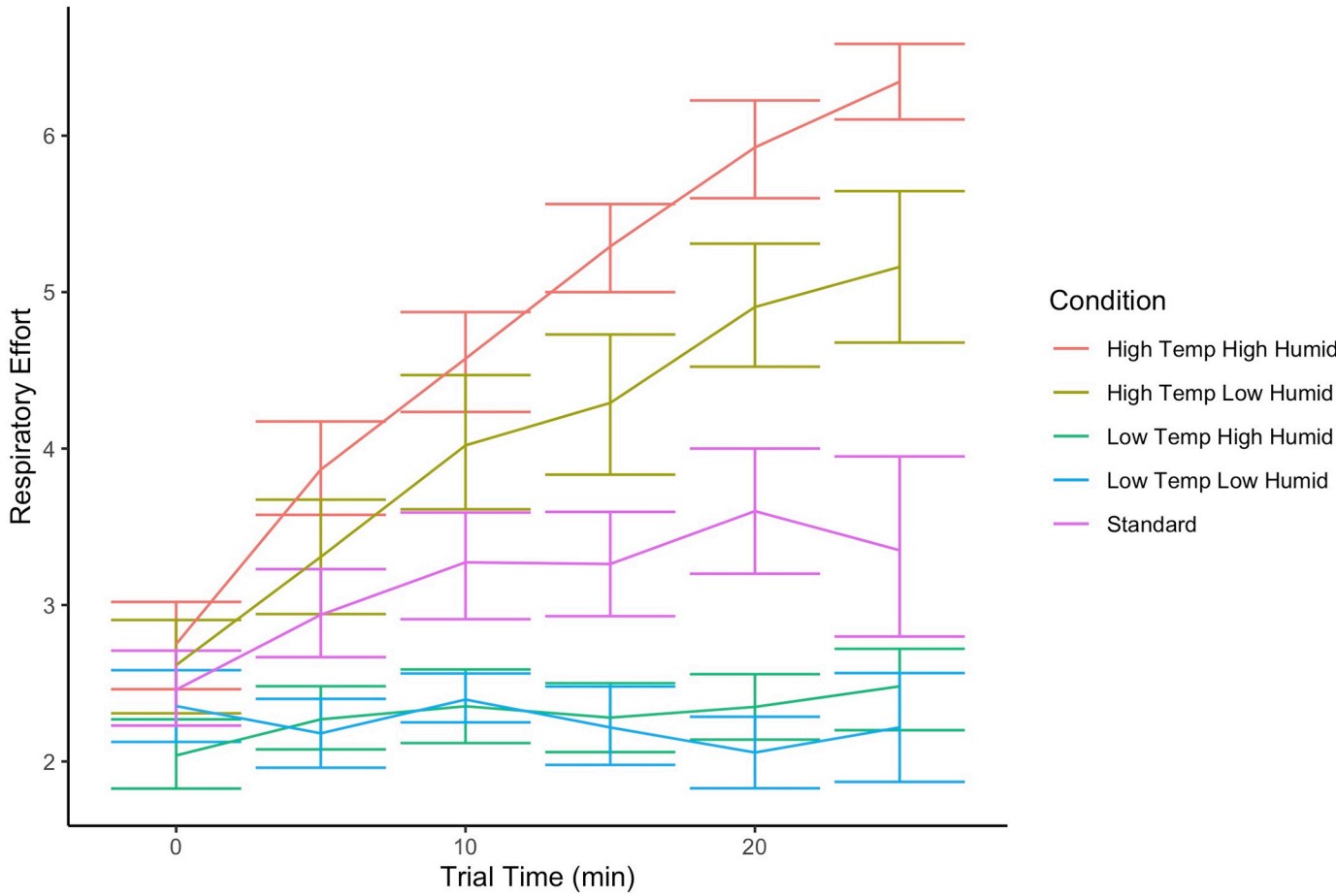

**Fig 9. Respiratory effort ratings during the threshold assessments.** Error bars show the boot strap estimated 95% confidence intervals.

## Discussion

In this study we established differences in odor sensitivity to varying energetic materials and changes in sensitivity due to extreme environmental conditions. Detection sensitivity limits were lowest (optimal) for canines for each odor at standard temperature and humidity conditions (22°C and 50% RH). Canines did not perform statistically better in any extreme environmental condition in comparison to standard conditions. Decrements in detection limits, were observed in each target odor for at least one of the extreme environmental conditions. High temperature and high humidity caused decrements in performance for SP and AN. High temperature low humidity caused decrements for AN, SP and a trend for C4. Low temperature high humidity led to decrements for C4 and SP. Low temperature low humidity only caused a decrement for SP.

No statistically significant effects between conditions were observed for TNT. This may reflect a restriction of range effect, dogs' threshold for TNT was near the highest odor concentration that could be presented by the air dilution olfactometer, even under standard conditions. Further, it was seen that the mean threshold of TNT across all 8 dogs was between the highest concentration and the second concentration presented to the dogs. Therefore there was a restriction in the decrement that could be observed because dogs' sensitivity limits were

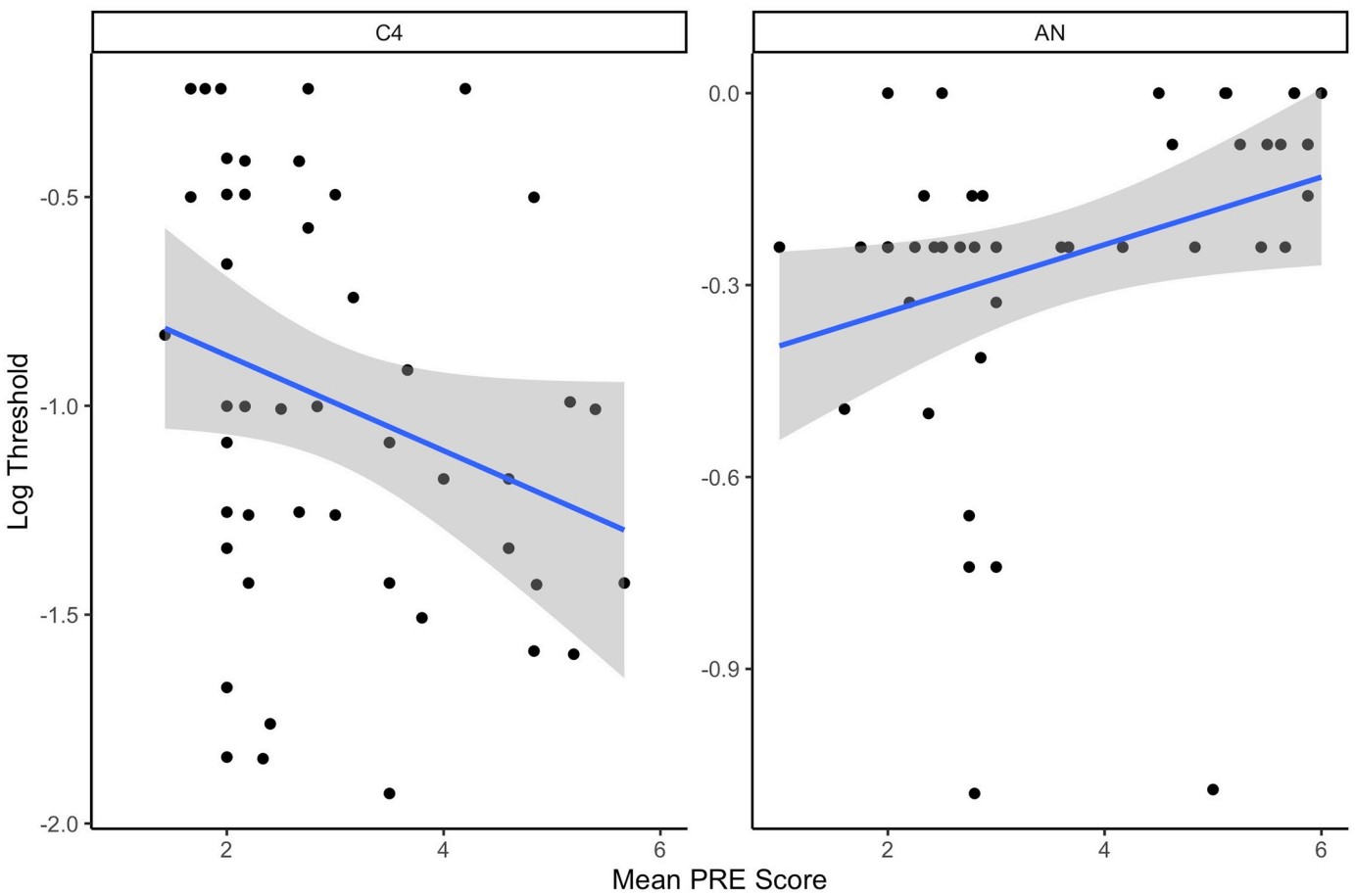

**Fig 10. Relationship between mean RE and log threshold.**

already at or near the highest concentrations that could be presented by the air dilution olfactometer.

Another unexpected finding is that dogs performed remarkably well in the high temperature and high humidity condition for C4. This appears to be driven largely by two dogs that started in the cold conditions and worked up to the hot conditions, whereas dogs that started in the hot conditions showed more of the expected decrement. C4 testing happened to coincide with ambient low humidity conditions in the region that caused more environmental variation when opening and closing the chamber to enter and begin testing. Retrospective analysis of the room conditions showed that although sessions started at the correct initial temperature and humidity conditions (high temperature and humidity), the temperature and relative humidity levels fell below expected levels after opening the chamber and starting testing in 7 of 16 testing sessions for C4, which did not occur for the other energetics (AN, SP, TNT). This may in part have led to the better than expected performance in this condition. In a follow-up study, Kane et al., (2024) demonstrated that dogs do show a performance decrement in the high temperature and high humidity condition for C4, highlighting that the observed drop in temperature may have been the reason for the lack of effect observed here [36].

There were interesting relationships in the measured physiological data and detection sensitivity. Subcutaneous temperature measurements were highest in the hot and humid conditions

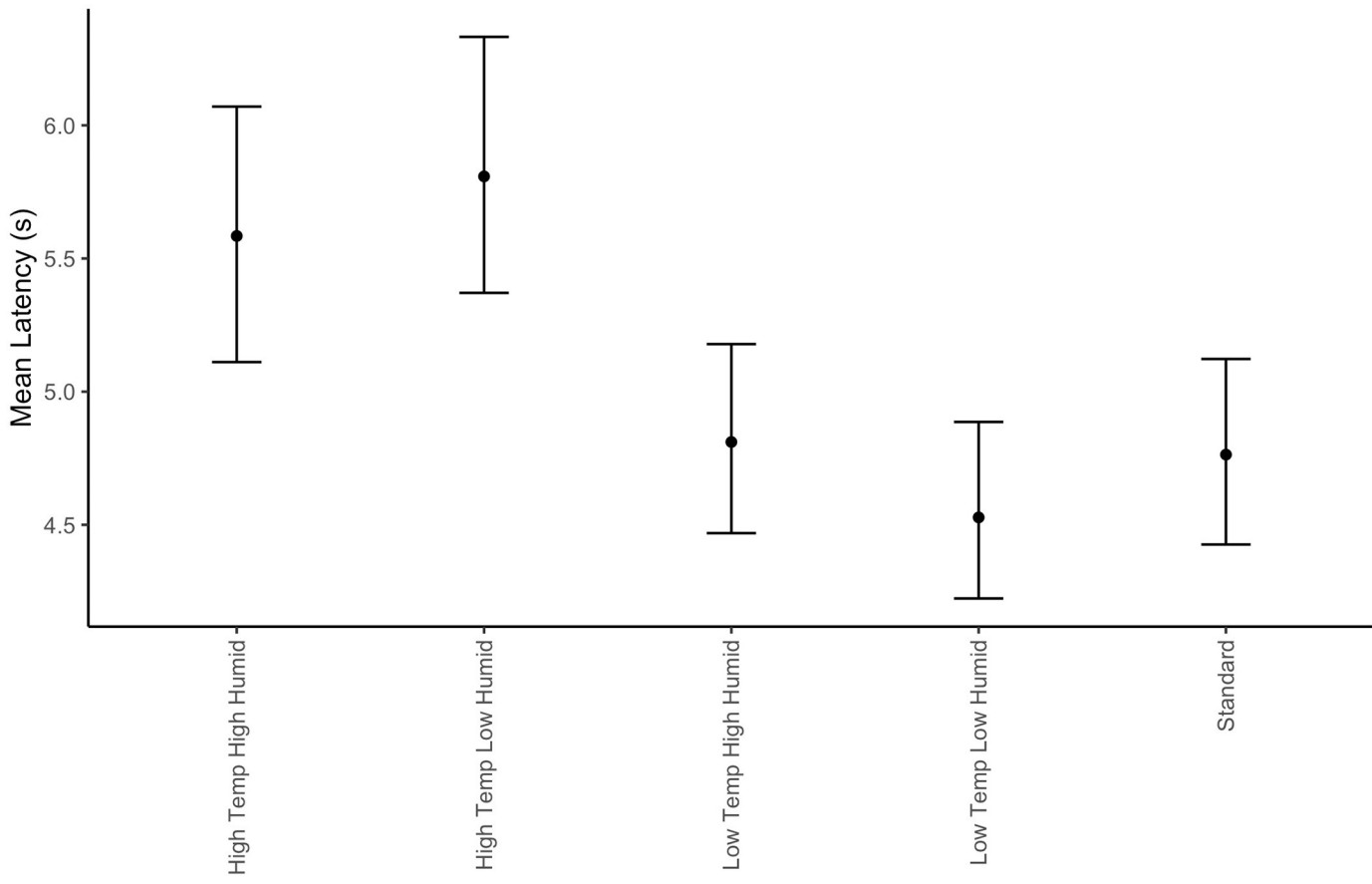

**Fig 11. Latency to sample odor port by environmental condition.** Error bars show the 95% bootstrap estimated confidence intervals.

in comparison to all other conditions; this indicates heat stress is at greatest risk in these environmental conditions. Dogs' mean subcutaneous temperature predicted detection performance for SP and AN, indicating that detection performance declined with greater increases in temperature. For example, for SP, a 1°C increase in subcutaneous temperature was associated with a 0.19 change in log threshold, reflecting a potentially important impact of canine subcutaneous temperature on performance. In the future it would be worth exploring how subcutaneous temperature and rectal, or core body temperature were related in this type of detection task study.

Respiratory effort (RE) also increased most in high temperature and high humidity conditions, followed by high temperature low humidity and standard conditions. RE was little impacted in the cold conditions. There is also some conflicting evidence of the effect of RE on threshold, such that overall, increases did lead to poorer detection limits for AN, which is to be expected based on prior work showing that respiratory effort can lead to poorer detection [25]. Notably, however, there was an opposite effect for C4. As RE scores increased from 1 to 6, it appears that there was a 1 log reduction in threshold. To better understand this discrepancy, further data is necessary.

Latency to initialize a trial also increased in both high temperature conditions compared to standard and cold conditions. Furthermore, longer latency to initiate a trial was associated with poorer detection thresholds. This suggests that lags in the dogs working behavior could

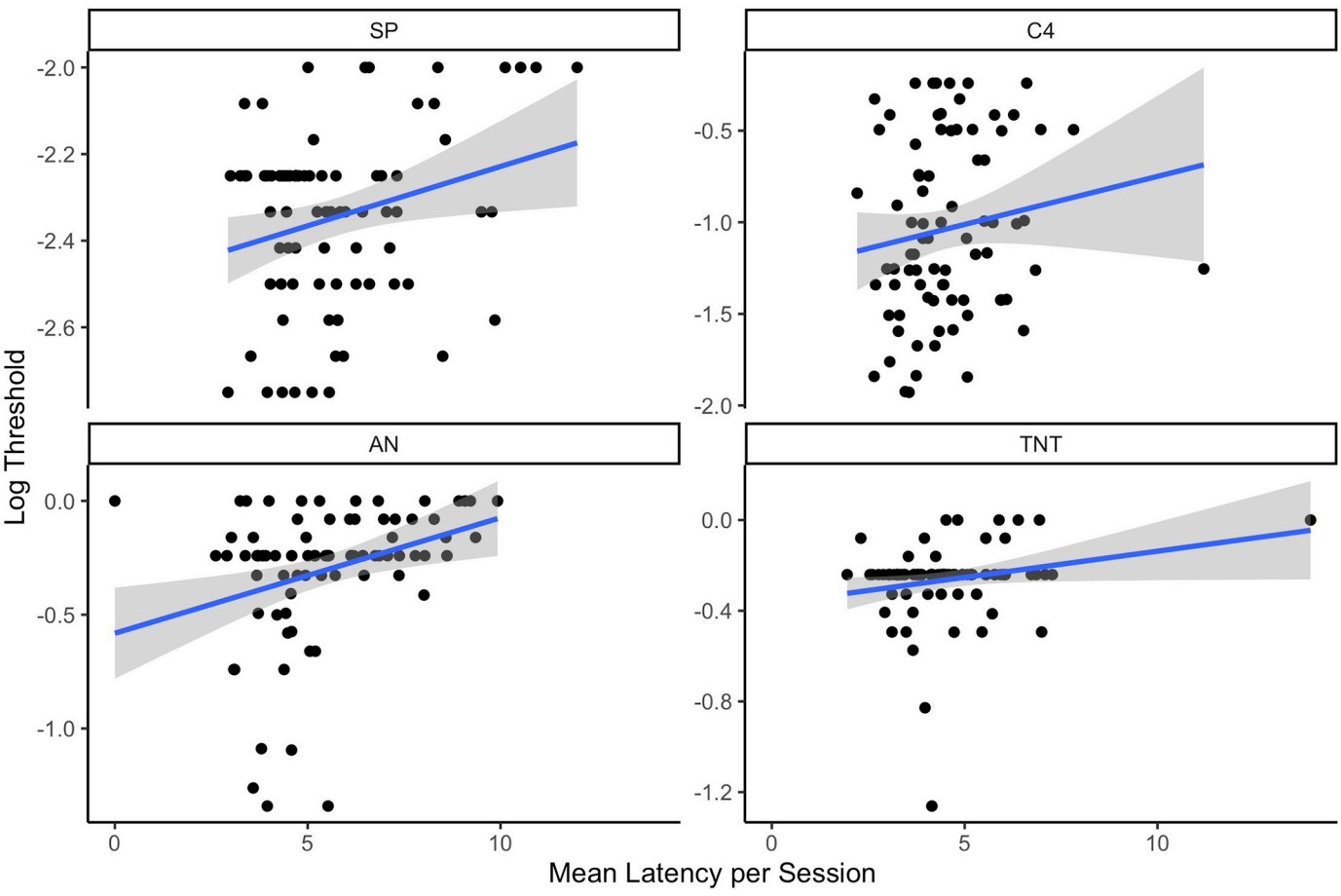

**Fig 12. Relationship between mean latency to initiate a trial in a session with overall threshold.**

be an important early cue that detection sensitivity may be poorer, or at least non-optimal. Latency to begin a search task, however, is not a common research topic and the field would benefit substantially from examining this in an operational context. A useful follow up experiment would evaluate to what degree the latency effect leads to more global decrements in canine performance (such as with obedience vs. a detection task). This would help elucidate whether the latency effect reflects a more generalized behavioral decrement or an olfactory specific decrement.

There are important considerations for the present study. First, odor concentration was maintained and controlled across environmental conditions with an air dilution olfactometer.

In operational searches, the odor source itself will be exposed to the extreme conditions and vary odor availability. This may change the effect observed such that although canine detection limits may change in high temperature and high humidity conditions, changes in odor availability may compensate minimizing overall performance decrements. Similarly, in cold conditions, even if canine detection capability remains similar, lower odor availability may change performance. Thus, additional evaluation in which odor source is allowed to change with environmental conditions is an important next step and this work is underway. As a follow up study to the present work the relationship between odor availability, environmental condition and canine threshold was examined Kane et al., (2024). Additionally this study examined how

effects of environment can be mitigated with acclimatization training in adverse environmental conditions [36].

Additionally, it should be noted that dogs were removed from the testing area based on our criterion for welfare considerations, or testing reached 40 minutes. Overall, across all dogs in all conditions and odors, dogs were removed early from 57 sessions out of 320 total sessions. However, performance prior to removal was poor, with a mean accuracy of 47%, indicating performance prior to early termination was below that expected by chance alone (50%). Importantly, only one dog was removed from one session during standard conditions, while the remaining 56 early termination sessions occurred during testing in extreme environmental conditions. Thus, early removal from testing sessions highlights the inability of the dogs to perform the working task better than chance, due to influences from the environmental conditions.

## Conclusion

Overall, this research demonstrates important considerations for working dogs. Environmental conditions had a negative impact on odor sensitivity for each odor tested in one or more of the conditions. This indicates that working dogs may face decrements in performance when working in varying environmental conditions. Lastly, physiological changes and changes in latency are indicators of possible decrements in odor sensitivity. Results do suggest that subcutaneous temperature, latency, and environmental conditions can be important predictors of changes in canine detection threshold (sensitivity) limits.

## Supporting information

**S1 Table. A RE scale rating system described by DeChant et al. (submitted) was used to determine the respiration effort of each participant during the five second clip for each five-minute interval.** Scoring for this scale ranges from 0 to 10. Zero being defined as resting and breathing is an involuntary process, and ten being very heavy breathing and the chest and abdomen are expanding and collapsing violently.
(DOCX)

**S1 Data. Raw data across all sessions compiled into one file.**
(CSV)

**S2 Data. Raw temperature data from each session.**
(CSV)

**S3 Data.**
(XLSX)

**S1 Code. Code in R used for all analyses and graphs presented.**
(CSV)

## Acknowledgments

We would like to thank Dr. Edgar O. Aviles-Rosa and Avery Bramlett for their help with data collection and helping with the care of the dogs. Further, we would like to thank the undergraduate students at the canine olfaction lab for providing care for the dogs involved in this project. Lastly, we would like to acknowledge Dr. Dillon Huff for providing helpful background information and samples used in this project.

## Author Contributions

**Conceptualization:** Paola A. Prada-Tiedemann, Nathaniel J. Hall.

**Data curation:** Lauren S. Fernandez, Sarah A. Kane, Nathaniel J. Hall.

**Formal analysis:** Lauren S. Fernandez, Sarah A. Kane, Nathaniel J. Hall.

**Funding acquisition:** Paola A. Prada-Tiedemann, Nathaniel J. Hall.

**Investigation:** Lauren S. Fernandez, Sarah A. Kane, Nathaniel J. Hall.

**Methodology:** Mallory T. DeChant, Nathaniel J. Hall.

**Project administration:** Paola A. Prada-Tiedemann, Nathaniel J. Hall.

**Resources:** Paola A. Prada-Tiedemann, Nathaniel J. Hall.

**Software:** Nathaniel J. Hall.

**Supervision:** Paola A. Prada-Tiedemann, Nathaniel J. Hall.

**Validation:** Paola A. Prada-Tiedemann.

**Visualization:** Lauren S. Fernandez, Sarah A. Kane, Nathaniel J. Hall.

**Writing – original draft:** Lauren S. Fernandez.

**Writing – review & editing:** Sarah A. Kane, Mallory T. DeChant, Paola A. Prada-Tiedemann, Nathaniel J. Hall.

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
