## [Decision Letter · Decision Letter 0]

28 Nov 2023

PONE-D-23-31665Environmental effects on explosive detection threshold of domestic dogsPLOS ONE

Dear Dr. Kane,

Thank you for submitting your manuscript to PLOS ONE. After careful consideration, we feel that it has merit but does not fully meet PLOS ONE’s publication criteria as it currently stands. Therefore, we invite you to submit a revised version of the manuscript that addresses the points raised during the review process.

As suggested by the reviewers, I am inviting you to revise the manuscript and submit the revised version for consideration.

We look forward to receiving your revised manuscript.

Kind regards,

Sankarganesh Devaraj

Academic Editor

PLOS ONE

3. We are unable to open your Supporting Information file [Analysis_Plos1_ptI .qmd]. Please kindly revise as necessary and re-upload.

Reviewers' comments:

Reviewer's Responses to Questions

**Comments to the Author**

1. Is the manuscript technically sound, and do the data support the conclusions?

Reviewer #1: Yes

Reviewer #2: Yes

2. Has the statistical analysis been performed appropriately and rigorously? 

Reviewer #1: Yes

Reviewer #2: Yes

3. Have the authors made all data underlying the findings in their manuscript fully available?

Reviewer #1: Yes

Reviewer #2: Yes

4. Is the manuscript presented in an intelligible fashion and written in standard English?

Reviewer #1: Yes

Reviewer #2: Yes

5. Review Comments to the Author

Reviewer #1: I find this paper as an interesting publication well written and presenting well-performed experiments. The methods are well described and are adequate to the subject. The number of animals used could be bigger, however, but in that kind of study usually a similar or even smaller amount of animals used to be used. The experimental group consisting of two breeds also seems to be even, which may be important in the context of the type of fur and sensitivity to environmental factors tested in this experiment. Results are clearly presented and come from the well-planned and performed experiment. The discussion is well led and the authors present interesting and important presenting also considerations on the impact of changing environmental conditions on behaviour, the "availability" of odors and not only the ability of dogs as detectors.

Below I attaching some doubts, questions and suggestions to the text and I will be grateful to the authors for their responses to the following comments.

Line 116 118 Interestingly, it should be noted that previous work has seen that dogs are able to detect target odors with no decline in performance while working in a variety of extreme 1conditions [30–32]

From other side there are papers presenting opposite opinion. Kokocińska et al reported numerous of papers indicating that humidity has been found to be an important factor influencing olfactory skills in dogs. In this case increased humidity can even improve canine olfactory skills, as if mentioned in a few papers cited in mentioned review of Kokocińka

Line 140-148 Due to greater heat stress, there is a potential that detection dogs may experience a decrease in detection accuracy while working in hot and humid environments. It has been found that landmine detection dogs had a significant decrease in detection accuracy after the working environment experienced heavy rain, ultimately increasing the humidity [48]. However, it should also be considered that the heavy rain potentially disrupted the soil containing the odor by washing the odorant away. But further data in this study showed that dogs had a decrease in detection accuracy early in the morning when the humidity was the highest and had an increase in detection success as humidity decreased throughout the day [48]. This supports the idea that an increase in humidity hinders an EDD’s ability to perform detection tasks

In my opinion, it would be worth to mention another opinion in this matter. In the mentioned review authors reported that: With regard to the effect of environmental conditions, humidity has been found to be an important factor in improving olfactory skills in dogs, probably due to improved nasal humidity and odorant trapping [Jenkins, E.K.; DeChant, M.T.; Perry, E.B. When the Nose Doesn’t Know: Canine Olfactory Function Associated with Health, Management, and Potential Links to Microbiota. Front. Veter. Sci. 2018, 5, 56.]

]. Moreover, according to Gutzwiller [Gutzwiller, K.J. Minimizing Dog-Induced Biases in Game Bird Research. Wildl. Soc. Bull. (1973–2006) 1990, 18, 351–356. [Google Scholar]

], increased humidity could be responsible for increased odor intensity, positively influencing the tracking efficiency of dogs. This phenomenon is also thought to be true in the context of semiochemical substance detection [Majumder, S.S.; Bhadra, A. When Love Is in the Air: Understanding Why Dogs Tend to Mate when It Rains. PLoS ONE 2015, 10, e0143501. [Google Scholar] [CrossRef] [PubMed][Green Version]

].

If authors do not agree with this thesis it would be worth discussing it anyway.

Line 182 All dogs were previously eliminated from working dog program.

What was the reason for elimination of those dogs from the working dog program?

Line 209 HVAC

please expand the abbreviation. I understand that it is Heating, Ventilation, Air Conditioning, but all abbreviations in the text supposed to be explained.

Line 232 Due to the variation of volatility of each material, flow rate used for dilution was dependent on odor

Can you explain if it was calculated somehow to propose that kind of dilution or maybe it was just established by the authors.

Line 279 a photo presenting dogs working is maybe not necessary but could be a useful addition that allows the reader to better understand the course of the experiment, not so much to imagine it

I have also a question about the latency. If I understand well, dogs in suboptimal conditions start working with lags and have a poorer results. Could it be connected with the fact that dogs sometimes don’t want to work ( have a worse day for work). Bad environmental conditions could influence the general involvement of the dogs and that could influence the detection results. I mean all other requested tasks could also be realized poorer ( obedience aspects for example). What do you think about this- could it be the reason for the worse final result of work?

Reviewer #2: Line 89: Suggest replace “sensitivity” with “detectability”. Vapor pressure does not affect olfactory sensitivity it affects detection. Also line 109, 118-119, 175 and 225 as well as additional occasions. For contrast, the term “sensitivity” used appropriately in line 324.

Line 95: Does “brand” really change vapor pressure of energetic material? Brand may change the chemicals present, but not the vapor pressure of the chemical.

Lines 96-100: This discussion needs some further background provided for the reader that is not well versed about energetic materials. First, TNT and DNT should be spelled out upon this or prior use of acronym as well as other acronyms, although in case of RDX, suggest may put chemical name in parentheses vs. typical way acronym is introduced. I think it should be noted that DNT is a constituent of the vapor from TNT. I don’t follow well the phrase “This fluctuation is also seen with RDX (a component of C4)” what fluctuation – between RDX and other energetic materials – way written could be construed as fluctuation in vapor pressure of RDX.

Line 104: Explosives contain other chemicals often due to purposeful addition of other materials to render them more user friendly in addition to contaminants and degradation products. Such purposeful additions are more important to detection by dogs because they are consistent.

Line 107: Table title references vapor pressure of explosive materials; while this is okay by itself, the term “energetic materials” has been used earlier in manuscript and I think that keeping the distinction between energetic chemicals and an explosive material produced using this chemical would be helpful throughout discussion to this point in manuscript. Indeed, of course, TNT is TNT and AN is not energetic in and of itself. Perhaps use of some explanation like “explosive materials are composed of energetic chemicals or some material in combination are energetic compounds. Also, note in abstract (line 30-31, AN is listed as an “energetic” material. I think some clearer description is warranted here for readers that do not have a background regarding energetic chemicals and explosive materials. Also line 397 and 400.

Line 131: Term “manipulate” here seems awkward.

Line 153: Grammar change – “…in which working dogs are tasked to search”

Line 153: Throughout manuscript there is a style issue of using a lot of transitional “further”, furthermore. For example in this line “Further, it has also been shown…” is wordy and not efficient. This could be written as “Divero et al. reported search and rescue dogs acclimated to working in -8.5C to -10.4C completing search tasks without signs of stress of fatigue [50].”

Line 157: “maybe” should be “may be”.

Line 158: suggest change “establish” to “investigate”.

Line 161: The phrase “It should be noted” and similar is overused throughout manuscript. If it is a reference to relevant previous studies, then the authors including it supposes it should be noted. In this line, perhaps consider “However, previous work found that dogs working in cold and standard conditions had the lowest threshold for methyl benzoate detection.” Also, this sentence does not contain a reference – was this previous unpublished work of the authors?

Line 252-254, perhaps noted elsewhere and can be ascertained by the description of olfactometer function , but suggest add that on non-odor trials the final air flow was the same as the total air flow of the diluted explosive (odor) trials.

Line 311. I do not believe that acronym PRE (assume “perceived respiratory effort” was defined earlier

Lines 348 – 357: Not clear about non-odor, blank, trials or were there NO non-odor (diluent only) trials during testing?

Lines 423~427 and table 7: Recommend that t-ratio comparison be described and explained in text.

Lines 447 – 467: It would have been useful to have a comparison of the sensor body temperature and that from another measurement (e.g., rectal) as a way to evaluate the sensor performance.

Line 545: Term “search performance” used for, what I believe is the first time. Suggest not using this term, particularly for first time in manuscript here. Environmental impact on “search performance” suggests searching and task was a fixed-sampling position activity that may not be considered a “search performance”.

Lines 610-616: This is an interesting finding, but it would be made more useful to reader if the numbers of standard sessions from which dogs were removed was noted. Perhaps reader is to assume these were all in the more extreme environmental conditions.

Lines 620 – 629: Suggest some discussion by authors about acclimation of dogs to environmental conditions and what impact it may have on changes seen in sensitivity due to environment in present experiment.

6. PLOS authors have the option to publish the peer review history of their article (what does this mean?). If published, this will include your full peer review and any attached files.

Reviewer #1: No

Reviewer #2: **Yes: **Paul Waggoner

---

## [Author Response · Author response to Decision Letter 0]

18 Apr 2024

Dear Reviewers, 

Thank you for your comments on our manuscript “Environmental effects on explosive detection threshold of domestic dogs.” Please see our responses below. 

Reviewer #1: 

I find this paper as an interesting publication well written and presenting well-performed experiments. The methods are well described and are adequate to the subject. The number of animals used could be bigger, however, but in that kind of study usually a similar or even smaller amount of animals used to be used. The experimental group consisting of two breeds also seems to be even, which may be important in the context of the type of fur and sensitivity to environmental factors tested in this experiment. Results are clearly presented and come from the well-planned and performed experiment. The discussion is well led and the authors present interesting and important presenting also considerations on the impact of changing environmental conditions on behaviour, the "availability" of odors and not only the ability of dogs as detectors.

Thank you for your thoughtful comments. 

Below I attaching some doubts, questions and suggestions to the text and I will be grateful to the authors for their responses to the following comments.

Line 116 118 Interestingly, it should be noted that previous work has seen that dogs are able to detect target odors with no decline in performance while working in a variety of extreme 1conditions [30–32]

From other side there are papers presenting opposite opinion. Kokocińska et al reported numerous of papers indicating that humidity has been found to be an important factor influencing olfactory skills in dogs. In this case increased humidity can even improve canine olfactory skills, as if mentioned in a few papers cited in mentioned review of Kokocińka

Thank you for your comment, we have addressed this oversight in line 153. 

Line 140-148 Due to greater heat stress, there is a potential that detection dogs may experience a decrease in detection accuracy while working in hot and humid environments. It has been found that landmine detection dogs had a significant decrease in detection accuracy after the working environment experienced heavy rain, ultimately increasing the humidity [48]. However, it should also be considered that the heavy rain potentially disrupted the soil containing the odor by washing the odorant away. But further data in this study showed that dogs had a decrease in detection accuracy early in the morning when the humidity was the highest and had an increase in detection success as humidity decreased throughout the day [48]. This supports the idea that an increase in humidity hinders an EDD’s ability to perform detection tasks

In my opinion, it would be worth to mention another opinion in this matter. In the mentioned review authors reported that: With regard to the effect of environmental conditions, humidity has been found to be an important factor in improving olfactory skills in dogs, probably due to improved nasal humidity and odorant trapping [Jenkins, E.K.; DeChant, M.T.; Perry, E.B. When the Nose Doesn’t Know: Canine Olfactory Function Associated with Health, Management, and Potential Links to Microbiota. Front. Veter. Sci. 2018, 5, 56.]

]. Moreover, according to Gutzwiller [Gutzwiller, K.J. Minimizing Dog-Induced Biases in Game Bird Research. Wildl. Soc. Bull. (1973–2006) 1990, 18, 351–356. [Google Scholar]

], increased humidity could be responsible for increased odor intensity, positively influencing the tracking efficiency of dogs. This phenomenon is also thought to be true in the context of semiochemical substance detection [Majumder, S.S.; Bhadra, A. When Love Is in the Air: Understanding Why Dogs Tend to Mate when It Rains. PLoS ONE 2015, 10, e0143501. [Google Scholar] [CrossRef] [PubMed][Green Version]

].

If authors do not agree with this thesis it would be worth discussing it anyway.

Thank you for providing additional citations to add nuance to our arguments. Addressed in lines 128-131, and noted below:

 This supports the idea that an increases in humidity hinder an EDD’s ability to perform detection tasks. However, a literature review by Jenkins et al., 2018 noted that a moist environment is crucial for olfactory perception, indicating the humidity may aid in olfaction [38]. Previous work with hunting canines has supports that increased humidity impacts tracking time [39].

Line 182 All dogs were previously eliminated from working dog program.

What was the reason for elimination of those dogs from the working dog program?

Description added :

“Dogs varied in their reasons for removal from their prior working program including preference for food vs. toy rewards, distractibility or unknown reasons.”

Line 209 HVAC please expand the abbreviation. I understand that it is Heating, Ventilation, Air Conditioning, but all abbreviations in the text supposed to be explained.

Good catch, acronym explained 267-268.

Line 232 Due to the variation of volatility of each material, flow rate used for dilution was dependent on odor. Can you explain if it was calculated somehow to propose that kind of dilution or maybe it was just established by the authors.

Details on flow rate determination have been added to the manuscript:

“The flow rate determined the highest range of concentration presented. During initial training, if dogs struggled to detect the 0.01 dilutions >85% accuracy, the total flow was set to 3 SLPM to allow for a higher range of concentrations.”

Line 279 a photo presenting dogs working is maybe not necessary but could be a useful addition that allows the reader to better understand the course of the experiment, not so much to imagine it

 Photo included in supplemental materials. Text included in line 709:

A photo of Rocket holding a freeze-nose hold alert in the olfactometer is included in supplemental materials. He is wearing all sensors mentioned above. 

I have also a question about the latency. If I understand well, dogs in suboptimal conditions start working with lags and have a poorer results. Could it be connected with the fact that dogs sometimes don’t want to work ( have a worse day for work). Bad environmental conditions could influence the general involvement of the dogs and that could influence the detection results. I mean all other requested tasks could also be realized poorer ( obedience aspects for example). What do you think about this- could it be the reason for the worse final result of work?

Yes, we agree completely. We have edited the discussion to hopefully clarify that this latency effect may suggest a generalized behavioral decrement and suggest future work looking at other behaviors. We have added the following: 

“A useful follow up experiment would evaluate to what degree the latency effect leads to more global decrements in canine performance (such as with obedience vs. a detection task). This would help elucidate whether the latency effect reflects a more generalized behavioral decrement or an olfactory specific decrement.”

Reviewer #2:

Line 89: Suggest replace “sensitivity” with “detectability”. Vapor pressure does not affect olfactory sensitivity it affects detection. Also line 109, 118-119, 175 and 225 as well as additional occasions. For contrast, the term “sensitivity” used appropriately in line 324.

Thank you, this was fixed. 

Line 95: Does “brand” really change vapor pressure of energetic material? Brand may change the chemicals present, but not the vapor pressure of the chemical.

Fixed in line 101.

Lines 96-100: This discussion needs some further background provided for the reader that is not well versed about energetic materials. First, TNT and DNT should be spelled out upon this or prior use of acronym as well as other acronyms, although in case of RDX, suggest may put chemical name in parentheses vs. typical way acronym is introduced. I think it should be noted that DNT is a constituent of the vapor from TNT. I don’t follow well the phrase “This fluctuation is also seen with RDX (a component of C4)” what fluctuation – between RDX and other energetic materials – way written could be construed as fluctuation in vapor pressure of RDX.

Thank you for these suggestions. We have done substantial re-writing of the introduction to address these unclarities and noted stylistic issues below. 

Line 104: Explosives contain other chemicals often due to purposeful addition of other materials to render them more user friendly in addition to contaminants and degradation products. Such purposeful additions are more important to detection by dogs because they are consistent.

Thank you for this note, we have added a comment on this in lines 127-129.

Line 107: Table title references vapor pressure of explosive materials; while this is okay by itself, the term “energetic materials” has been used earlier in manuscript and I think that keeping the distinction between energetic chemicals and an explosive material produced using this chemical would be helpful throughout discussion to this point in manuscript. Indeed, of course, TNT is TNT and AN is not energetic in and of itself. Perhaps use of some explanation like “explosive materials are composed of energetic chemicals or some material in combination are energetic compounds. Also, note in abstract (line 30-31, AN is listed as an “energetic” material. I think some clearer description is warranted here for readers that do not have a background regarding energetic chemicals and explosive materials. Also line 397 and 400.

Thank you for this clarification. We have revised the manuscript throughout for clarity in terminology and use energetic materials throughout. 

Line 131: Term “manipulate” here seems awkward.

Addressed line 166

Line 153: Grammar change – “…in which working dogs are tasked to search”

Thank you, this change was made in line 163. 

Line 153: Throughout manuscript there is a style issue of using a lot of transitional “further”, furthermore. For example in this line “Further, it has also been shown…” is wordy and not efficient. This could be written as “Divero et al. reported search and rescue dogs acclimated to working in -8.5C to -10.4C completing search tasks without signs of stress of fatigue [50].”

We have substantially edited the introduction and the writing throughout to improve quality. 

Line 157: “maybe” should be “may be”.

Addressed line 214. 

Line 158: suggest change “establish” to “investigate”.

Thank you, addressed in line 215. 

Line 161: The phrase “It should be noted” and similar is overused throughout manuscript. If it is a reference to relevant previous studies, then the authors including it supposes it should be noted. In this line, perhaps consider “However, previous work found that dogs working in cold and standard conditions had the lowest threshold for methyl benzoate detection.” Also, this sentence does not contain a reference – was this previous unpublished work of the authors?

Citation oversight fixed. We have re-worked the writing style.

Line 252-254, perhaps noted elsewhere and can be ascertained by the description of olfactometer function , but suggest add that on non-odor trials the final air flow was the same as the total air flow of the diluted explosive (odor) trials.

A very important clarification point, added in lines 367-368. 

Line 311. I do not believe that acronym PRE (assume “perceived respiratory effort” was defined earlier

Acronym corrected 389-390. 

Lines 348 – 357: Not clear about non-odor, blank, trials or were there NO non-odor (diluent only) trials during testing?

We have clarified that there were no-odor (clean air) trials presented with equal probability to odor present trials. 

Lines 423~427 and table 7: Recommend that t-ratio comparison be described and explained in text.

Thank you for this note, issue has been addressed in lines 662-667. 

Lines 447 – 467: It would have been useful to have a comparison of the sensor body temperature and that from another measurement (e.g., rectal) as a way to evaluate the sensor performance.

Yes, this would have been an interesting area to explore, a note on this added in lines 691-692 in the discussion. 

Line 545: Term “search performance” used for, what I believe is the first time. Suggest not using this term, particularly for first time in manuscript here. Environmental impact on “search performance” suggests searching and task was a fixed-sampling position activity that may not be considered a “search performance”.

Thank you, changed to “detection performance”

Lines 610-616: This is an interesting finding, but it would be made more useful to reader if the numbers of standard sessions from which dogs were removed was noted. Perhaps reader is to assume these were all in the more extreme environmental conditions.

Addressed 731-732, we agree this is an important clarification for the reader. We have added: “Importantly, only one dog was removed from one session during all testing sessions in standard conditions, while the remaining 56 early termination sessions occurred during testing in extreme environmental conditions.”

Lines 620 – 629: Suggest some discussion by authors about acclimation of dogs to environmental conditions and what impact it may have on changes seen in sensitivity due to environment in present experiment.

 We did a continuation of this study which examined the relationship between acclimatization and detection performance. Results from that study, and citation for it added at the end of the discussion; lines 748-750.

---

## [Decision Letter · Decision Letter 1]

25 Jun 2024

Environmental effects on explosive detection threshold of domestic dogs

PONE-D-23-31665R1

Dear Dr. Kane,

We’re pleased to inform you that your manuscript has been judged scientifically suitable for publication and will be formally accepted for publication once it meets all outstanding technical requirements.

Kind regards,

Sankarganesh Devaraj

Academic Editor

PLOS ONE

Additional Editor Comments (optional):

The comments of the reviewers have been addressed appropriately. Therefore, I recommend the manuscript for publication in its current form.

Reviewers' comments:

Reviewer's Responses to Questions

**Comments to the Author**

1. If the authors have adequately addressed your comments raised in a previous round of review and you feel that this manuscript is now acceptable for publication, you may indicate that here to bypass the “Comments to the Author” section, enter your conflict of interest statement in the “Confidential to Editor” section, and submit your "Accept" recommendation.

Reviewer #1: All comments have been addressed

2. Is the manuscript technically sound, and do the data support the conclusions?

Reviewer #1: Yes

3. Has the statistical analysis been performed appropriately and rigorously? 

Reviewer #1: Yes

4. Have the authors made all data underlying the findings in their manuscript fully available?

Reviewer #1: Yes

5. Is the manuscript presented in an intelligible fashion and written in standard English?

Reviewer #1: Yes

6. Review Comments to the Author

Reviewer #1: In my opinion, the MS improved after changes made according the reviewers suggestioons and in this form it can be accepted, although I think I still notice a small error in the description of the tables: Line:

134 ( 230 in corrected version) Detailed descriptions of each participant can be found in Table 2.

As I can see, description of the participants are available in the table 1 not 2 which presents “Description of odorants”

7. PLOS authors have the option to publish the peer review history of their article (what does this mean?). If published, this will include your full peer review and any attached files.

Reviewer #1: No

---

## [Editor Report · Acceptance letter]

2 Sep 2024

PONE-D-23-31665R1 

PLOS ONE

Dear Dr. Kane, 

I'm pleased to inform you that your manuscript has been deemed suitable for publication in PLOS ONE. Congratulations! Your manuscript is now being handed over to our production team.

Kind regards, 

on behalf of

Dr. Sankarganesh Devaraj 

Academic Editor

PLOS ONE